# q-means: A quantum algorithm for unsupervised machine learning

**Iordanis Kerenidis**
CNRS, IRIF, Université Paris Diderot, Paris, France
`jkeren@irif.fr`

**Jonas Landman**
CNRS, IRIF, Universiteé Paris Diderot, Paris, France
Ecole Polytechnique, Palaiseau, France. `landman@irif.fr`

**Alessandro Luongo**
CNRS, IRIF, Universiteé Paris Diderot, Paris, France
Atos Quantum Lab - Les Clayes-sous-Bois, France
`aluongo@irif.fr`

**Anupam Prakash**
CNRS, IRIF, Universiteé Paris Diderot, Paris, France
`anupam.prakash@irif.fr`

## Abstract

Quantum information is a promising new paradigm for fast computations that can provide substantial speedups for many algorithms we use today. Among them, quantum machine learning is one of the most exciting applications of quantum computers. In this paper, we introduce $q$-means, a new quantum algorithm for clustering. It is a quantum version of a robust $k$-means algorithm, with similar convergence and precision guarantees. We also design a method to pick the initial centroids equivalent to the classical $k$-means$++$ method. Our algorithm provides currently an exponential speedup in the number of points of the dataset, compared to the classical $k$-means algorithm. We also detail the running time of $q$-means when applied to *well-clusterable* datasets. We provide a detailed runtime analysis and numerical simulations for specific datasets. Along with the algorithm, the theorems and tools introduced in this paper can be reused for various applications in quantum machine learning.

## 1   Introduction

As the amount of data generated in our society is expected to grow faster than the growth in our computational capabilities, more powerful ways of processing information are needed. Quantum computation uses the fundamental properties of quantum physics to redefine the way computers create and manipulate information. These properties imply a radically new way of computing, using *qubits* instead of bits, and give the possibility of obtaining quantum algorithms that could be substantially faster than classical algorithms. In recent years, there have been proposals for quantum machine learning algorithms that have the potential to offer considerable speedups over the corresponding classical algorithms, either exponential or large polynomial speedups [28, 23, 22, 8, 27, 3]. Of course, in order to translate such theoretical results into advantages for real-world use cases one

would need both more advanced quantum hardware, which might be still years away, but also a close collaboration between the classical and quantum machine learning communities in order to better understand when and how quantum algorithms can be used as a powerful tool within the larger machine learning framework. In most of these quantum machine learning applications, there are some common algorithmic primitives that are used to build the algorithms. For instance, quantum procedures for linear algebra (matrix multiplication, inversion, and projections in sub-eigenspaces of matrices) have been used for recommendation systems or dimensionality reduction techniques [23, 21, 28]. Second, the ability to estimate distances between quantum states, for example through the SWAP test, has been used for supervised or unsupervised learning [27, 36]. We note that most of these procedures can be used either with quantum data or they need quantum access to the classical data, which can be achieved by storing the data in specific data structures like a QRAM (Quantum Random Access Memory).

In this paper, we propose *q-means*, a quantum algorithm for *clustering*, which can be viewed as a quantum analogue to the classical $k$-means algorithm. Since quantum computation is not deterministic and is also prone to noise, quantum machine learning has to incorporate a certain level of randomness. Therefore it is more precise to present $q$-means as a quantum equivalent of the $\delta$-$k$-means algorithm, which is a version of $k$-means with noise, introduced in this paper. We provide an analysis to show that the output of $q$-means is consistent with the classical $\delta$-$k$-means algorithm and further that the running time depends poly-logarithmically on the number of elements in the dataset. The simplest version of the $k$-means algorithm runs in time $O(ndtk)$, where $n$ is the number of elements in the training set, $d$ is the number of features, $t$ is the number of iterations, and $k$ is the number of classes.

Previous work in quantum clustering exists [27, 31]. In our opinion, the biggest limitation of previous works is that they do not allow to retrieve the classical fitted model out of the computation (i.e. in our case are the $k$ centroids). Quantum-inspired classical algorithm for clustering exsists [20, 9]. These achieve polylogarithmic scaling in terms of numer of elements in the dataset, but are of a much higher polynomial degree with respect to other parameters (like condition number, rank and error). Algorithms based on these techniques have been implemented and benchmarked on real dataset with discouraging results [5], therefore we don't expect these results to change the set problem where quantum computers are expected to get an edge of advantage over classical computation. A complete review of previous works is presented in Supplementary Material, Section A.1

### 1.1 The $k$-means and $\delta$-$k$-means Algorithms

The input for the $k$-means algorithm [29] is a dataset $V$ of vectors $v_i \in \mathbb{R}^d$ for $i \in [N]$. These points must be partitioned in $k$ subsets according to a similarity measure, e.g. the Euclidean distance. The output of the $k$-means algorithm is a list of $k$ cluster centers, which are called *centroids*. At iteration $t$, we denote the $k$ clusters by the sets $C_j^t$ for $j \in [k]$, and each corresponding centroid by the vector $c_j^t$. Each data point $v_i$ is assigned to one cluster $C_j^t$. Let $d(v_i, c_j^t)$ be the Euclidean distance between vectors $v_i$ and $c_j^t$. The algorithm starts by selecting $k$ initial centroids and then alternates between two steps: (i) assign each $v_i$ a label $\ell(v_i)^t$ corresponding to the closest centroid, that is $\ell(v_i)^t = \text{argmin}_{j \in [k]}(d(v_i, c_j^t))$. (ii) update the centroids with the following rule: $c_j^{t+1} = \frac{1}{|C_j^t|} \sum_{v_i \in C_j^t} v_i$. We say that we have converged if for a small threshold $\tau$ we have $\frac{1}{k} \sum_{j=1}^k d(c_j^t, c_j^{t-1}) \leqslant \tau$. We now introduce $\delta$-$k$-means, that can be thought as a robust version of the $k$-means algorithm in which we introduce some noise parametrized by $\delta > 0$. The noise affects the algorithm in both steps of the $k$-means algorithm: label assignment and centroid estimation. As we will see in this work, $q$-means is the quantum analog of $\delta$-$k$-means, due to the noise and non deterministic character of quantum computations. Let $c_i^*$ be the closest centroid to the data point $v_i$. In the assignment step, instead of choosing deterministically the label corresponding to the closest centroid, one label is randomly assigned among the followhing set:

$$L_\delta(v_i) = \{p : |d^2(c_i^*, v_i) - d^2(c_p, v_i)| \leq \delta \} \tag{1}$$

Second, we add $\delta/2$ noise during the calculation of the centroid. Let $\mathcal{C}_j^{t+1}$ be the set of points which have been labeled by $j$ in the previous step. For $\delta$-k-means we pick a centroid $c_j^{t+1}$ with the property $d(c_j^{t+1}, \frac{1}{|\mathcal{C}_j^{t+1}|} \sum_{v_i \in \mathcal{C}_j^{t+1}} v_i) < \frac{\delta}{2}$. We simulate this by adding small Gaussian noise to the centroid.

Let us add two remarks on the $\delta$-$k$-means. First, for a *well-clusterable* dataset (see Section 1.4) and for a small $\delta$, the number of vectors on the boundary that risk to be misclassified in each step, that is the vectors for which $|L_\delta(v_i)| > 1$, is typically much smaller compared to the vectors that are close to a unique centroid. Second, we also increase by $\delta/2$ the convergence threshold from the $k$-means algorithm. All in all, the $\delta$-$k$-means algorithm finds a clustering that is robust when the data points and the centroids are perturbed with noise of magnitude $O(\delta)$. Our numerical simulations show that the performance of the $\delta$-$k$-means algorithm is similar to the $k$-means algorithm for small enough $\delta$'s.

## 1.2 Quantum Preliminaries

We assume a basic understanding of quantum computing, we recommend Nielsen and Chuang [30] for an introduction to the subject. A vector $v \in \mathbb{R}^d$ is encoded into a quantum state $|v\rangle$ defined as $|v\rangle = \frac{1}{\|v\|} \sum_{m \in [d]} v_m |m\rangle$, where $|m\rangle$ represents $e_m$, the $m^{th}$ vector in the standard basis. A quantum circuit or algorithm consists of unitary logic gates or measurements, and can be applied to a superposition of quantum states. We will assume at some steps that the data matrices $V$ (datapoints) and $C^t$ (centroids at step $t$) are stored in suitable QRAM data structures which are described in [23]. Important quantum subroutines and theorems for this work are described in Supplementary Material, Section A.3.

## 1.3 Our Results

We define and analyse a new quantum algorithm for clustering, the $q$-means algorithm, whose running time provides substantial savings, especially for the case of large data sets, and whose performance is similar to that of the classical $\delta$-$k$-means algorithm - a robust version of the $k$-means algorithm we defined in this work - meaning that with high probability the clusters that the $q$-means algorithm outputs are also possible outputs for the $\delta$-$k$-means.

The $q$-means algorithm combines most of the advantages that quantum machine learning algorithms can offer for clustering. First, the running time is poly-logarithmic in the number of elements of the dataset and depends only linearly on the dimension of the feature space. Second, $q$-means returns explicit classical descriptions of the cluster centroids that are obtained by the $\delta$-$k$-means algorithm. As the algorithm outputs a classical description of the centroids, it is possible to use them in further (classical or quantum) algorithms, unlike previous works on quantum $k$-means [27] that outputs quantum states corresponding to the centroids. We start by providing a worst case analysis of the running time of each step of our algorithm. The running time parameters include the maximum norm of the dataset, the condition number and a parameter $\mu$ of the data point matrix (see definition in Theorem 3.1). While different than the classical case, these aspects are common in quantum computing [22], where the magnitude or the rank of the data point matrix can impact the efficiency of the algorithm itself. Note that with $\widetilde{O}$ we hide polylogarithmic factors.

**Result 1.** *Given dataset $V \in \mathbb{R}^{N \times d}$ stored in QRAM, the q-means algorithm outputs with high probability centroids $c_1, \cdots, c_k$ that are consistent with an output of the $\delta$-k-means algorithm in time $\widetilde{O}\left(kd\frac{\eta}{\delta^2}\kappa(V)(\mu(V) + k\frac{\eta}{\delta}) + k^2\frac{\eta^{1.5}}{\delta^2}\kappa(V)\mu(V)\right)$ per iteration, where $\kappa(V)$ is the condition number, $\mu(V)$ is a parameter that appears in quantum linear algebra procedures and $1 \leq \|v_i\|^2 \leq \eta$.*

We also provide a specific running time analysis for a natural notion of *well-clusterable* datasets, given in the following section. See Theorem 3.2 for formal proof.

**Result 2.** *Given a well-clusterable dataset $V \in \mathbb{R}^{N \times d}$ stored in QRAM, the q-means algorithm outputs with high probability $k$ centroids $c_1, \cdots, c_k$ that are consistent with the output of the $\delta$-k-means algorithm in time $\widetilde{O}\left(k^2 d\frac{\eta^{2.5}}{\delta^3} + k^{2.5}\frac{\eta^2}{\delta^3}\right)$ per iteration, where $1 \leq \|v_i\|^2 \leq \eta$.*

The parameter $\delta$ (which plays the same role as in the $\delta$-$k$-means) is expected to be a large enough constant that depends on the data, and the parameter $\eta$ is again expected to be a small constant for datasets whose data points have roughly the same norm. In high level, our algorithm is quadratic on the number of clusters, linear in the dimension of points and only polylogarithmic in the number of data points. We present extensive simulations for different datasets and found that the number of iterations is practically the same as in the $k$-means, and the $\delta$-$k$-means algorithm achieves an accuracy similar to the $k$-means algorithm, see Section 4.

## 1.4 Modelling Well-Clusterable Datasets

Without loss of generality we consider in the remaining of the paper that the dataset $V$ is normalized so that for all $i \in [N]$, we have $1 \leq \|v_i\|$, and we define the parameter $\eta = \max_i \|v_i\|^2$. We will also assume that the number $k$ is the "right" number of clusters, meaning that we assume each cluster has at least some $\Omega(N/k)$ data points. We now propose a simple notion of *well-clusterable* dataset. The definition aims to capture some properties that we can expect from datasets that can be clustered efficiently using a $k$-means algorithm. Note that we do not need this assumption for our general $q$-means algorithm, but in this model we can provide tighter bounds for its running time. Our definition of a well-clusterable dataset shares some similarity with the models made in [12], [25] but they remain specific for our current problem.

**Definition 1** (Well-clusterable dataset). *A data matrix $V \in \mathbb{R}^{N \times d}$ with rows $v_i \in \mathbb{R}^d, i \in [N]$ is said to be well-clusterable if there exist constants $\xi, \beta > 0$, $\lambda \in [0, 1]$, $\eta \leq 1$, and cluster centroids $c_i$ for $i \in [k]$ such that:*

*- (separation of cluster centroids): $d(c_i, c_j) \geq \xi \quad \forall i, j \in [k]$*

*- (proximity to cluster centroid): At least $\lambda N$ points $v_i$ in the dataset satisfy $d(v_i, c_{l(v_i)}) \leq \beta$ where $c_{l(v_i)}$ is the centroid nearest to $v_i$.*

*- (Intra-cluster smaller than inter-cluster square distances): The following inequality is satisfied $4\sqrt{\eta}\sqrt{\lambda\beta^2 + (1-\lambda)4\eta} \leq \xi^2 - 2\sqrt{\eta}\beta$.*

Intuitively, the assumptions guarantee that most of the data can be easily assigned to one of $k$ clusters, since these points are close to the centroids, and the centroids are sufficiently far from each other. The exact inequality comes from the error analysis, but in spirit it says that the intra-cluster distance must be suficiently smaller than the inter-cluster distance. A series of four claims are detailed in the Supplementary Material, Section A.2 that provide mathematical properties of well-clusterable datasets, and will be used in the proofs on the running time of the $q$-means applied to well-clusterable datasets.

An overview of $q$-means algorithm is given as Algorithm 1.

## 2 The $q$-means Algorithm

At a high level, the $q$-means algorithm follows the same steps as the classical $k$-means algorithm, where we now use quantum subroutines for distance estimation, finding the minimum value among a set of elements, matrix multiplication for obtaining the new centroids as quantum states, and efficient tomography. First, we pick $k$ random centroids, or we use our initialization procedure $q$-means++, an efficient quantum equivalent of $k$-means++ (see Section 2.1). Then, in Steps 1 and 2 all data points are assigned to clusters in superposition and not one after the other, and in Steps 3 and 4 we update the centroids of the clusters. The process is repeated until convergence.

**Step 1: Centroid Distance Estimation** The first step of the algorithm estimates the square distance between all data points and centroids using a quantum procedure. From $|i\rangle |j\rangle |0\rangle$, we create the state $|i\rangle |j\rangle \left(\frac{1}{2} |0\rangle (|v_i\rangle + |c_j\rangle) + \frac{1}{2} |1\rangle (|v_i\rangle - |c_j\rangle)\right)$. Since the probability of measuring 1 on the third register is proportional to $\langle v_i | c_j \rangle$, we can use the Amplitude Estimation circuit to extract this value in another quantum register. Several copies of this register can be taken to compute a median estimation that boost the probability of success. More details are provided in the Supplementary Material, Section A.4.1. The distance estimation becomes very efficient when we have quantum access to the vectors and the centroids by querying the state preparation oracles with the QRAM $|i\rangle |0\rangle \mapsto |i\rangle |v_i\rangle$, and $|j\rangle |0\rangle \mapsto |j\rangle |c_j\rangle$ in time $T = O(\log nd)$, as well as querying the norm of these vectors. As quantum states have unit norm, we need to multiply their inner products by the real norms $\|v_i\| \|c_j\|$. If we have an absolute error $\epsilon$ for the square distance estimation of the normalized vectors, then the final error is of the order of $\epsilon \|v_i\| \|c_j\|$. These computations are performed in superposition over all point indices $|i\rangle$ and for a tensor product of all centroid indices $|j\rangle$ at the same time. It leads to the distance estimation theorem corresponding to Step 1 of $q$-means algorithm. We develop its proof in the Supplementary Material, Section A.4.1.

**Theorem 2.1** (Centroid Distance estimation). *Let a data matrix $V \in \mathbb{R}^{N \times d}$ and a centroid matrix $C \in \mathbb{R}^{k \times d}$ be stored in QRAM, such that the following unitaries $|i\rangle |0\rangle \mapsto |i\rangle |v_i\rangle$, and $|j\rangle |0\rangle \mapsto$*

$|j\rangle\,|c_j\rangle$ *can be performed in time* $O(\log(Nd))$ *and the norms of the vectors are known. For any* $\Delta > 0$ *and* $\epsilon_1 > 0$, *there exists a quantum algorithm that, given the state* $\frac{1}{\sqrt{N}}\sum_{i=1}^{N}|i\rangle \otimes_{j\in[k]}(|j\rangle\,|0\rangle)$, *performs the mapping to*

$$\frac{1}{\sqrt{N}}\sum_{i=1}^{N}|i\rangle \otimes_{j\in[k]}(|j\rangle\,|\overline{d^2(v_i,c_j)}\rangle), \tag{2}$$

*where* $|\overline{d^2(v_i,c_j)} - d^2(v_i,c_j)| \leqslant \epsilon_1$ *with probability at least* $1 - 2\Delta$, *in time* $\widetilde{O}\left(k\frac{\eta\log(\Delta^{-1})}{\epsilon_1}\right)$ *where* $\eta = \max_i(\|v_i\|^2)$.

**Step 2: Cluster Assignment**   At the end of step 1, we have coherently estimated the square distance between each point in the dataset and the $k$ centroids in separate registers. We can now select the index $j$ that corresponds to the centroid closest to the given data point, written as $\ell(v_i) = \mathrm{argmin}_{j\in[k]}(d(v_i,c_j))$. As the square is a monotone function, we do not need to compute the square root of the distance in order to find $\ell(v_i)$.

**Lemma 2.2** (Circuit for finding the minimum). *Given* $k$ *different* $\log p$-bit registers $\otimes_{j\in[k]}|a_j\rangle$, *there is a quantum circuit* $U_{min}$ *that maps* $(\otimes_{j\in[k]}|a_j\rangle)|0\rangle \rightarrow (\otimes_{j\in[k]}|a_j\rangle)|\mathrm{argmin}(a_j)\rangle$ *in time* $O(k\log p)$.

*Proof.* We append an additional register for the result that is initialized to $|1\rangle$. We then repeat the following operation for $2 \leq j \leq k$, we compare registers 1 and $j$, if the value in register $j$ is smaller we swap registers 1 and $j$ and update the result register to $j$. The cost of the procedure is $O(k\log p)$. $\qquad\square$

The cost of finding the minimum is $\widetilde{O}(k)$ in step 2 of the $q$-means algorithm, while we also need to uncompute the distances by repeating Step 1. Once we apply the minimum finding Lemma 2.2 and undo the computation we obtain the state

$$|\psi^t\rangle := \frac{1}{\sqrt{N}}\sum_{i=1}^{N}|i\rangle\,|\ell^t(v_i)\rangle. \tag{3}$$

In high level Steps 1 and 2 have assigned labels to all data points in superposition. Note that this state does not allow us to read out all possible labels, but it contains exactly the information we need in order to estimate the new centroids in the following step.

**Step 3: Centroid state creation**   The previous step gave us the state $|\psi^t\rangle = \frac{1}{\sqrt{N}}\sum_{i=1}^{N}|i\rangle\,|\ell^t(v_i)\rangle$. The first register of this state stores the index of the data points while the second register stores the label for the data point in the current iteration. Given these states, we need to find the new centroids $|c_j^{t+1}\rangle$, which are the barycenters of the data points having the same label. Let $\chi_j^t \in \mathbb{R}^N$ be the characteristic vector for cluster $j \in [k]$ at iteration $t$ scaled to unit $\ell_1$ norm, that is $(\chi_j^t)_i = \frac{1}{|C_j^t|}$ if $i \in \mathcal{C}_j$ and 0 if $i \notin \mathcal{C}_j$. The creation of the quantum states corresponding to the centroids is based on the following simple claim.

**Claim 2.3.** *Let* $\chi_j^t \in \mathbb{R}^N$ *be the scaled characteristic vector for* $\mathcal{C}_j$ *at iteration* $t$ *and* $V \in \mathbb{R}^{N\times d}$ *be the data matrix, then* $c_j^{t+1} = V^T\chi_j^t$.

The above claim allows us to compute the updated centroids $c_j^{t+1}$ using quantum linear algebra operations. In fact, the state $|\psi^t\rangle$ can be written as a weighted superposition of the characteristic vectors of the clusters.

$$|\psi^t\rangle = \sum_{j=1}^{k}\sqrt{\frac{|C_j|}{N}}\left(\frac{1}{\sqrt{|C_j|}}\sum_{i\in\mathcal{C}_j}|i\rangle\right)|j\rangle = \sum_{j=1}^{k}\sqrt{\frac{|C_j|}{N}}\,|\chi_j^t\rangle\,|j\rangle \tag{4}$$

We can then measure the label register $|j\rangle$. The running time of this step is derived from Theorem A.8, in Supplementary Material, where the time to prepare the state $|\chi_j^t\rangle$ is the time of Steps 1 and 2. Note that we do not have to add an extra $k$ factor due to the sampling, since we can run the matrix multiplication procedures in parallel for all $j$ so that every time we measure a random $|\chi_j^t\rangle$ we perform one more step of the corresponding matrix multiplication. Assuming that all clusters have size $\Omega(N/k)$ we will have an extra factor of $O(\log k)$ in the running time by a standard coupon collector argument.

**Step 4: Centroid update**   In Step 4, we need to go from quantum states corresponding to the centroids, to a classical description of the centroids in order to perform the update step. For this, we will apply the vector state tomography algorithm, stated in Theorem A.9, in Supplementary Material, on the states $|c_j^{t+1}\rangle$ that we create in Step 3. Note that for each $j \in [k]$ we will need to invoke the unitary that creates the states $|c_j^{t+1}\rangle$ a total of $O(\frac{d \log d}{\epsilon_4^2})$ times for achieving $\||c_j\rangle - |\overline{c_j}\rangle\| < \epsilon_4$. Hence, for performing the tomography of all clusters, we will invoke the unitary $O(\frac{k(\log k)d(\log d)}{\epsilon_4^2})$ times where the $O(k \log k)$ term is the time to get a copy of each centroid state. The vector state tomography gives us a classical estimate of the unit norm centroids within error $\epsilon_4$, that is $\||c_j\rangle - |\overline{c_j}\rangle\| < \epsilon_4$. Using the approximation of the norms $\|c_j\|$ with relative error $\epsilon_3$ from Step 3, we can combine these estimates to recover the centroids as vectors. The analysis is described in the following claim, whose proof can be found in the Supplementary Material:

**Claim 2.4.** *Let $\epsilon_4$ be the error we commit in estimating $|c_j\rangle$ such that $\||c_j\rangle - |\overline{c_j}\rangle\| < \epsilon_4$, and $\epsilon_3$ the error we commit in the estimating the norms, $|\,\|c_j\| - \overline{\|c_j\|}\,| \le \epsilon_3 \|c_j\|$. Then $\|\overline{c_j} - c_j\| \le \sqrt{\eta}(\epsilon_3 + \epsilon_4) = \epsilon_{centroid}$.*

## 2.1   Initialization: $q$-means++

The $k$-means++ technique [6] is frequently used for initializing the classical $k$-means algorithm. The first centroid is chosen uniformly at random. We sample the next centroid from a probability distribution where the probability of sampling $v_i$ is proportional to the squared distance to the closest centroid. We add the sampled point to the list of the already chosen centroids, and repeat the procedure until $k$ centroids have been chosen. Note that when more than one centroids are already picked, the sampling probability is proportional to the squared distance to the closest centroid. In the Supplementary Material (Section A.5) we prove the following theorem:

**Theorem 2.5.** *Let the data matrix $V \in \mathbb{V} \in \mathbb{R}^{N \times d}$ be stored in the QRAM. There exists a quantum algorithm that returns a matrix $C \in \mathbb{R}^{k \times d}$ consistent with the centroids returned by the $k$-means++ initialization algorithm in time*

$$O\left(k^2 \frac{2\eta^{1.5}}{\epsilon_1 \sqrt{\mathbb{E}(d^2(v_i, v_j))}}\right), \tag{5}$$

*where $\mathbb{E}(d^2(v_i, v_j))$ is the average squared distance between two points of the dataset.*

# 3   Analysis

We provide our general theorem about the running time and accuracy of the $q$-means algorithm.

**Theorem 3.1** ($q$-means). *For a data matrix $V \in \mathbb{R}^{N \times d}$ and parameter $\delta > 0$, the q-means algorithm with high probability outputs centroids consistent with the classical $\delta$-k-means algorithm, in time $\widetilde{O}\left(kd\frac{\eta}{\delta^2}\kappa(V)(\mu(V) + k\frac{\eta}{\delta}) + k^2\frac{\eta^{1.5}}{\delta^2}\kappa(V)\mu(V)\right)$ per iteration, where $1 \le \|v_i\|^2 \le \eta$, $\kappa(V)$ is the condition number, and $\mu(V) = \min_{p \in \mathcal{P}}\left(\|V\|_F, \sqrt{s_{2p}(V)s_{2(1-p)}(V^T)}\right)$, where $\mathcal{P} \subset [0, 1]$ such that $|\mathcal{P}| = O(1)$ and $s_p(V) := \max_{i \in [N]} \|V_i\|_p^p$*

We prove this theorem in Sections 3.1 and 3.2 and then provide the running time of the algorithm for well-clusterable datasets as Theorem 3.2.

**Algorithm 1** $q$-means.

---

**Require:** Data matrix $V \in \mathbb{R}^{N \times d}$ stored in QRAM data structure. Precision parameters $\delta$ for $k$-means, error parameters $\epsilon_1$ for distance estimation, $\epsilon_2$ and $\epsilon_3$ for matrix multiplication and $\epsilon_4$ for tomography.

**Ensure:** Outputs vectors $c_1, c_2, \cdots, c_k \in \mathbb{R}^d$ that correspond to the centroids at the final step of the $\delta$-$k$-means algorithm.

Select $k$ initial centroids $c_1^0, \cdots, c_k^0$ and store them in QRAM data structure.
t=0
**repeat**

    **Step 1: Centroid Distance Estimation**
    Perform the mapping (Theorem 2.1)

$$\frac{1}{\sqrt{N}} \sum_{i=1}^{N} |i\rangle \otimes_{j \in [k]} |j\rangle |0\rangle \mapsto \frac{1}{\sqrt{N}} \sum_{i=1}^{N} |i\rangle \otimes_{j \in [k]} |j\rangle |\overline{d^2(v_i, c_j^t)}\rangle \tag{6}$$

    where $|\overline{d^2(v_i, c_j^t)} - d^2(v_i, c_j^t)| \leq \epsilon_1$.
    **Step 2: Cluster Assignment**
    Find the minimum distance among $\{d^2(v_i, c_j^t)\}_{j \in [k]}$ (Lemma 2.2), then uncompute Step 1 to create the superposition of all points and their labels

$$\frac{1}{\sqrt{N}} \sum_{i=1}^{N} |i\rangle \otimes_{j \in [k]} |j\rangle |\overline{d^2(v_i, c_j^t)}\rangle \mapsto \frac{1}{\sqrt{N}} \sum_{i=1}^{N} |i\rangle |\ell^t(v_i)\rangle \tag{7}$$

    **Step 3: Centroid states creation**

    **3.1** Measure the label register to obtain a state $|\chi_j^t\rangle = \frac{1}{\sqrt{|\mathcal{C}_j^t|}} \sum_{i \in \mathcal{C}_j^t} |i\rangle$, with prob. $\frac{|\mathcal{C}_j^t|}{N}$

    **3.2** Perform matrix multiplication with matrix $V^T$ and vector $|\chi_j^t\rangle$ to obtain the state $|c_j^{t+1}\rangle$ with error $\epsilon_2$, along with an estimation of $\left\|c_j^{t+1}\right\|$ with relative error $\epsilon_3$ (Theorem A.8).
    **Step 4: Centroid Update**
    **4.1** Perform tomography for the states $|c_j^{t+1}\rangle$ with precision $\epsilon_4$ using the operation from Steps 1-3 (Theorem A.9) and get a classical estimate $\overline{c}_j^{t+1}$ for the new centroids such that $|c_j^{t+1} - \overline{c}_j^{t+1}| \leq \sqrt{\eta}(\epsilon_3 + \epsilon_4) = \epsilon_{centroids}$
    **4.2** Update the QRAM data structure for the centroids with the new vectors $\overline{c}_0^{t+1} \cdots \overline{c}_k^{t+1}$.
    t=t+1
**until** convergence condition is satisfied.

---

### 3.1 Error analysis

In this section we determine the error parameters in the different steps of the quantum algorithm so that the quantum algorithm behaves the same as the classical $\delta$-$k$-means. More precisely, we will determine the values of the errors $\epsilon_1, \epsilon_2, \epsilon_3, \epsilon_4$ in terms of $\delta$ so that firstly, the cluster assignment of all data points made by the $q$-means algorithm is consistent with a classical run of the $\delta$-$k$-means algorithm, and also that the centroids computed by the $q$-means after each iteration are again consistent with centroids that can be returned by the $\delta$-$k$-means algorithm. The cluster assignment in $q$-means happens in two steps. The first step estimates the square distances between all points and all centroids. The error in this procedure is of the form: $|\overline{d^2(c_j, v_i)} - d^2(c_j, v_i)| < \epsilon_1$. for a point $v_i$ and a centroid $c_j$. The second step finds the minimum of these distances without adding any error. For the $q$-means to output a cluster assignment consistent with the $\delta$-$k$-means algorithm, we require that:

$$\forall j \in [k], \quad |\overline{d^2(c_j, v_i)} - d^2(c_j, v_i)| \leq \frac{\delta}{2} \tag{8}$$

which implies that no centroid with distance more than $\delta$ above the minimum distance can be chosen by the $q$-means algorithm as the label. Thus we need to take $\epsilon_1 < \delta/2$. After the cluster assignment of the $q$-means (which happens in superposition), we update the clusters, by first performing a matrix multiplication to create the centroid states and estimate their norms, and then a tomography to get

a classical description of the centroids. The error in this part is $\epsilon_{centroids}$, as defined in Claim 2.4, namely: $\|\overline{c}_j - c_j\| \le \epsilon_{centroid} = \sqrt{\eta}(\epsilon_3 + \epsilon_4)$.. Again, for ensuring that the $q$-means is consistent with the classical $\delta$-$k$-means algorithm we take $\epsilon_3 < \frac{\delta}{4\sqrt{\eta}}$ and $\epsilon_4 < \frac{\delta}{4\sqrt{\eta}}$. Note also that we have ignored the error $\epsilon_2$ that we can easily deal with since it only appears in a logarithmic factor.

## 3.2 Runtime analysis

As the classical algorithm, the runtime of $q$-means depends linearly on the number of iterations, so here we analyze the cost of a single step. The cost of tomography for the $k$ centroid vectors is $O(\frac{kd\log k\log d}{\epsilon_4{}^2})$ times the cost of preparation of a single centroid state $|c_j^t\rangle$. A single copy of $|c_j^t\rangle$ is prepared applying the matrix multiplication by $V^T$ procedure on the state $|\chi_j^t\rangle$ obtained using square distance estimation. The time required for preparing a single copy of $|c_j^t\rangle$ is $O(\kappa(V)(\mu(V) + T_\chi)\log(1/\epsilon_2))$ by Theorem A.8 where $T_\chi$ is the time for preparing $|\chi_j^t\rangle$. The time $T_\chi$ is $\widetilde{O}\left(\frac{k\eta\log(\Delta^{-1})\log(Nd)}{\epsilon_1}\right) = \widetilde{O}(\frac{k\eta}{\epsilon_1})$ by Theorem 2.1.

The cost of norm estimation for $k$ different centroids is independent of the tomography cost and is $\widetilde{O}(\frac{kT_\chi\kappa(V)\mu(V)}{\epsilon_3})$. Combining together all these costs and suppressing all the logarithmic factors we have a total running time of $\widetilde{O}\left(kd\frac{1}{\epsilon_4^2}\kappa(V)\left(\mu(V) + k\frac{\eta}{\epsilon_1}\right) + k^2\frac{\eta}{\epsilon_3\epsilon_1}\kappa(V)\mu(V)\right)$ The analysis in section 3.1 shows that we can take $\epsilon_1 = \delta/2$, $\epsilon_3 = \frac{\delta}{4\sqrt{\eta}}$ and $\epsilon_4 = \frac{\delta}{4\sqrt{\eta}}$. Substituting these values in the above running time, it follows that the running time of the $q$-means algorithm is

$$\widetilde{O}\left(kd\frac{\eta}{\delta^2}\kappa(V)\left(\mu(V) + k\frac{\eta}{\delta}\right) + k^2\frac{\eta^{1.5}}{\delta^2}\kappa(V)\mu(V)\right). \tag{9}$$

This completes the proof of Theorem 3.1. We next state our main result when applied to a well-clusterable dataset, as in Section 1.4.

**Theorem 3.2** ($q$-means on well-clusterable data). *For a well-clusterable dataset $V \in \mathbb{R}^{N\times d}$ stored in appropriate QRAM, the $q$-means algorithm returns with high probability the $k$ centroids consistently with the classical $\delta$-$k$-means algorithm for a constant $\delta$ in time $\widetilde{O}\left(k^2d\frac{\eta^{2.5}}{\delta^3} + k^{2.5}\frac{\eta^2}{\delta^3}\right)$ per iteration, for $1 \le \|v_i\|^2 \le \eta$.*

The proof of this Theorem is provided in Supplementary Material, Section A.4.3. At high level, we use claims to bound the parameters $\kappa(V)$ and $\mu(V)$ of well-clusterable datasets, as well as error parameters, thanks to the rank, singular values and distribution properties of such datasets.

Let us make some concluding remarks regarding the running time of $q$-means. For dataset where the number of points is much bigger compared to the other parameters, the running time for the $q$-means algorithm is an improvement compared to the classical $k$-means algorithm. For instance, for most problems in data analysis, $k$ is eventually small ($< 100$). The number of features $d \le N$ in most situations, and it can eventually be reduced by applying a quantum dimensionality reduction algorithm [21] first (which have running time poly-logarithmic in $d$). To sum up, $q$-means has the same output as the classical $\delta$-$k$-means algorithm (which is a robust version of k-means with similar running time and performance), it conserves the same number of iterations, but has a running time only poly-logarithmic in $N$, giving an exponential speedup with respect to the size of the dataset.

## 4 Simulations on real data

We would like to demonstrate that the quantum algorithm provides accurate classification results. However, since neither quantum simulators nor quantum computers large enough to test $q$-means are available currently, we tested the equivalent classical implementation of $\delta$-$k$-means, knowing that our quantum algorithms provides results consistent with the $\delta$-$k$-means algorithm. For implementing the $\delta$-$k$-means, we changed the assignment step of the $k$-means algorithm to select a random centroid among those that are $\delta$-close to the closest centroid and added $\delta/2$ error to the updated clusters. We benchmarked our $q$-means algorithm on two datasets: the well known MNIST dataset of handwritten digits and a synthetic dataset of gaussian clusters. To measure and compare the accuracy of our

clustering algorithm, we ran the $k$-means and the $\delta$-$k$-means algorithms for different values of $\delta$ on a training dataset and then we compared the accuracy of the classification on a test set, containing data points on which the algorithms have not been trained, using a number of widely-used performance measures. More experiments are provided in the Supplementary Material.

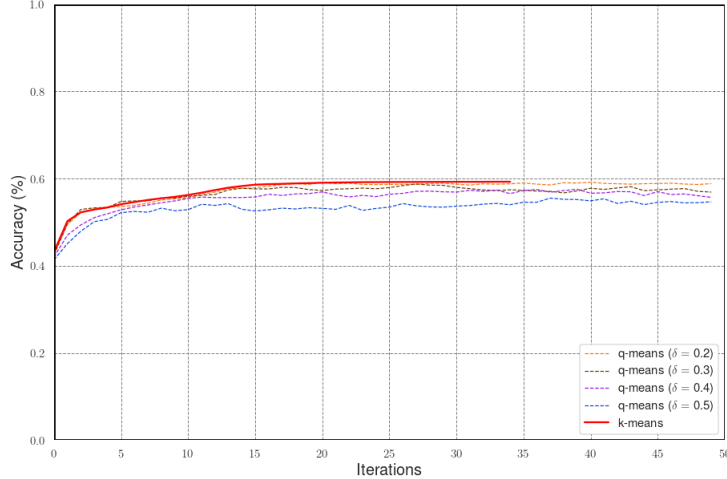

Figure 1: Accuracy evolution on the MNIST dataset under $k$-means and $q$-means ($\delta$-$k$-means) for $4$ different values of $\delta$. Data has been preprocessed by a PCA to 40 dimensions. All versions converge in the same number of steps, with a drop in the accuracy while $\delta$ increases. The apparent increase of steps until convergence is just due to the fact of the different stopping condition for $\delta$-$k$-means.

The MNIST dataset is composed of 60.000 handwritten digits as images of 28x28 pixels (784 dimensions). From this data we first performed some dimensionality reduction processing, then we normalized the data such that the minimum norm is one. Note that a quantum computer could also be used for dimensionality reduction algorithms like [28, 10]. As preprocessing, we first performed a Principal Component Analysis (PCA), retaining data projected in a subspace of dimension 40. After normalization, the value of $\eta$ was 8.25 (maximum norm of 2.87), and the condition number for the data matrix was 4.53. Figure 1 represents the evolution of the accuracy during the $k$-means and $\delta$-$k$-means for $4$ different values of $\delta$. In this numerical experiment, we can see that for values of the parameter $\eta/\delta$ of order 20, both $k$-means and $\delta$-$k$-means reached a similar accuracy in the same number of steps. Notice that the MNIST dataset, without other preprocessing than dimensionality reduction, is known not to be well-clusterable, hence the low accuracy reached. More experimental details are provided in Supplementary Material, Section A.7.

**Conclusions** In our experiments, the values of $\eta/\delta$ remained between 3 and 20. Moreover, the parameter $\eta = \max_i \|v_i\|^2$ provides a worst case guarantee for the algorithm. One can expect that the running time in practice will scale with the average square norm of the points. For the MNIST dataset after PCA, this value is 2.65 whereas $\eta = 8.3$. Our simulations show that the convergence rate of $\delta$-$k$-means is almost the same as the regular $k$-means algorithms even for large enough $\delta$. This provides evidence that the $q$-means algorithm will have as good performance as the classical $k$-means, with a running time that is significantly lower than that of the classical algorithms for large datasets.

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
