[Supplementary Material]

# A   q-means: Supplementary Meterial

## A.1   Related Work

Before providing the mathematical analysis, we discuss previous work on quantum unsupervised learning and clustering. Aimeur, Brassard and Gambs [2] gave two quantum algorithms for unsupervised learning using the amplification techniques from [13]. Specifically, they proposed an algorithm for clustering based on minimum spanning trees that runs in time $\Theta(N^{3/2})$ and a quantum algorithm for $k$-median (a problem related to k-means) algorithm with complexity time $O(N^{3/2}/\sqrt{k})$.

Lloyd, Mohseni and Rebentrost [27] proposed quantum $k$-means and nearest centroid algorithms using an efficient subroutine for quantum distance estimation assuming as we do quantum access to the data. Given a dataset of $N$ vectors in a feature space of dimension $d$, the running time of each iteration of the clustering algorithm (using a distance estimation procedure with error $\epsilon$) is $O\left(\frac{kN\log d}{\epsilon}\right)$ to produce the quantum state corresponding to the clusters. Note that the time is linear in the number of data points and it will be linear in the dimension of the vectors if the algorithm needs to output the classical description of the clusters.

In the same work, they also proposed an adiabatic algorithm for the assignment step of the $k$-means algorithm, that can potentially provide an exponential speedup in the number of data points as well, in the case the adiabatic algorithm performs exponentially better than the classical algorithm. The adiabatic algorithm is used in two places for this algorithm, the first to select the initial centroids, and the second to assign data points to the closest cluster. However, while arguments are given for its efficiency, it is left as an open problem to determine how well the adiabatic algorithm performs on average, both in terms of the quality of solution and the running time.

Wiebe, Kapoor and Svore [36] apply the minimum finding algorithm [13] to obtain nearest-neighbor methods for supervised and unsupervised learning. At a high level, they recovered a Grover-type quadratic speedup with respect to the number of elements in the dataset in finding the $k$ nearest neighbors of a vector. Otterbach et al. [32] performed clustering by exploiting a well-known reduction from clustering to the Maximum-Cut (MAXCUT) problem; the MAXCUT is then solved using QAOA, a quantum algorithm for performing approximate combinatorial optimization [14].

Let us remark on a recent breakthrough by Tang et al. [17, 35, 34], who proposed three classical machine learning algorithms obtained by dequantizing recommendation systems [23] and low rank linear system solvers. Like the quantum algorithms, the running time of these classical algorithms is $O(\text{poly}(k)\text{polylog}(mn))$, that is poly-logarithmic in the dimension of the dataset and polynomial in the rank. However, the polynomial dependence on the rank of the matrices is significantly worse than the quantum algorithms and in fact renders these classical algorithms highly impractical. For example, the new classical algorithm for stochastic regression inspired by the HHL algorithm [19] has a running time of $\widetilde{O}(\kappa^6 k^{16} \|A\|_F^6 /\epsilon^6)$, which is impractical even for a rank-10 matrix.

The extremely high dependence on the rank and the other parameters implies not only that the quantum algorithms are substantially faster (their dependence on the rank is sublinear!), but also that in practice there exist much faster classical algorithms for these problems. While the results of Tang et al. are based on the FKV methods [16], in classical linear algebra, algorithms based on the CUR decomposition that have a running time linear in the dimension and quadratic in the rank are preferred to the FKV methods [16, 11, 1]. For example, for the recommendation systems matrix of Amazon or Netflix, the dimension of the matrix is $10^6 \times 10^7$, while the rank is certainly not lower than 100. The dependence on the dimension and rank of the quantum algorithm in [23] is $O(\sqrt{k}\log(mn)) \approx O(10^2)$, of the classical CUR-based algorithm is $O(mk^2) \approx O(10^{11})$, while of the Tang algorithm is $O(k^{16}log(mn)) \approx O(10^{33})$.

It remains an open question to find classical algorithms for these machine learning problems that are poly-logarithmic in the dimension and are competitive with respect to the quantum or the classical algorithms for the same problems. This would involve using significantly different techniques than the ones presently used for these algorithms.

## A.2 Well-clusterable dataset model: Details

We refer to the Definiton 1 of Section 1.4 concerning well-clusterable datasets and provide some claims. Note that our $q$-means algorithm will provide good clustering whenever the classical $\delta$-$k$-means algorithm will, and not only for well-clusterable datasets. The reason for defining well-clusterable datasets is to make a mathematically rigorous running time analysis possible.

We now show that a well-clusterable dataset has a good rank-$k$ approximation where $k$ is the number of clusters. This result will later be used for giving tight upper bounds on the running time of the quantum algorithm for well-clusterable datasets. As we said, one can easily construct such datasets by picking $k$ well separated vectors to serve as cluster centers and then each point in the cluster is sampled from a Gaussian distribution with small variance centered on the centroid of the cluster.

We denote as $V_k$ the optimal rank $k$ approximation of $V$, that is $V_k = \sum_{i=0}^{k} \sigma_i u_i v_i^T$, where $u_i, v_i$ are the row and column singular vectors respectively and the sum is over the largest $k$ singular values $\sigma_i$. We denote as $V_{\geq \tau}$ the matrix $\sum_{i=0}^{\ell} \sigma_i u_i v_i^T$ where $\sigma_\ell$ is the smallest singular value which is greater than $\tau$.

**Claim A.1.** *Let $V_k$ be the optimal $k$-rank approximation for a well-clusterable data matrix $V$, then* $\|V - V_k\|_F^2 \leq (\lambda \beta^2 + (1 - \lambda)4\eta) \|V\|_F^2$.

*Proof.* Let $W \in \mathbb{R}^{N \times d}$ be the matrix with row $w_i = c_{l(v_i)}$, where $c_{l(v_i)}$ is the centroid closest to $v_i$. The matrix $W$ has rank at most $k$ as it has exactly $k$ distinct rows. As $V_k$ is the optimal rank-$k$ approximation to $V$, we have $\|V - V_k\|_F^2 \leq \|V - W\|_F^2$. It therefore suffices to upper bound $\|V - W\|_F^2$. Using the fact that $V$ is well-clusterable, we have

$$\|V - W\|_F^2 = \sum_{ij} (v_{ij} - w_{ij})^2$$

$$= \sum_i d(v_i, c_{l(v_i)})^2 \leq \lambda N \beta^2 + (1 - \lambda)N4\eta,$$

where we used Definition 1 to say that for a $\lambda N$ fraction of the points $d(v_i, c_{l(v_i)})^2 \leq \beta^2$ and for the remaining points $d(v_i, c_{l(v_i)})^2 \leq 4\eta$. Also, as all $v_i$ have norm at least 1 we have $N \leq \|V\|_F$, implying that $\|V - V_k\|^2 \leq \|V - W\|_F^2 \leq (\lambda \beta^2 + (1 - \lambda)4\eta) \|V\|_F^2$.

$\square$

The running time of the quantum linear algebra routines for the data matrix $V$ in Theorem A.8 depend on the parameters $\mu(V)$ and $\kappa(V)$. We establish bounds on both of these parameters using the fact that $V$ is well-clusterable

**Claim A.2.** *Let $V$ be a well-clusterable data matrix, then $\mu(V) := \frac{\|V\|_F}{\|V\|} = O(\sqrt{k})$.*

*Proof.* We show that when we rescale $V$ so that $\|V\| = 1$, then we have $\|V\|_F = O(\sqrt{k})$ for the rescaled matrix. From the triangle inequality we have that $\|V\|_F \leq \|V - V_k\|_F + \|V_k\|_F$. Using the fact that $\|V_k\|_F^2 = \sum_{i \in [k]} \sigma_i^2 \leq k$ and Claim A.1, we have,

$$\|V\|_F \leq \sqrt{(\lambda \beta^2 + (1 - \lambda)4\eta)} \|V\|_F + \sqrt{k}$$

Rearranging, we have that $\|V\|_F \leq \frac{\sqrt{k}}{1 - \sqrt{(\lambda \beta^2 + (1 - \lambda)4\eta)}} = O(\sqrt{k})$.

$\square$

We next show that if we use a condition threshold $\kappa_\tau(V)$ instead of the true condition number $\kappa(V)$, that is we consider the matrix $V_{\geq \tau} = \sum_{\sigma_i \geq \tau} \sigma_i u_i v_i^T$ by discarding the smaller singular values $\sigma_i < \tau$, the resulting matrix remains close to the original one, i.e. we have that $\|V - V_{\geq \tau}\|_F$ is bounded.

**Claim A.3.** *Let $V$ be a matrix with a rank-$k$ approximation given by $\|V - V_k\|_F \leq \epsilon' \|V\|_F$ and let $\tau = \frac{\epsilon_\tau}{\sqrt{k}} \|V\|_F$, then $\|V - V_{\geq \tau}\|_F \leq (\epsilon' + \epsilon_\tau) \|V\|_F$.*

*Proof.* Let $l$ be the smallest index such that $\sigma_l \geq \tau$, so that we have $\|V - V_{\geq\tau}\|_F = \|V - V_l\|_F$. We split the argument into two cases depending on whether $l$ is smaller or greater than $k$.

- If $l \geq k$ then $\|V - V_l\|_F \leq \|V - V_k\|_F \leq \epsilon' \|V\|_F$.

- If $l < k$ then, $\|V - V_l\|_F \leq \|V - V_k\|_F + \|V_k - V_l\|_F \leq \epsilon' \|V\|_F + \sqrt{\sum_{i=l+1}^{k} \sigma_i^2}$.

  As each $\sigma_i < \tau$ and the sum is over at most $k$ indices, we have the upper bound $(\epsilon' + \epsilon_\tau) \|V\|_F$.

$\square$

The reason we defined the notion of well-clusterable dataset is to be able to provide some strong guarantees for the clustering of most points in the dataset. Note that the clustering problem in the worst case is NP-hard and we only expect to have good results for datasets that have some good property. Intuitively, we should only expect $k$-means to work when the dataset can actually be clusterd in $k$ clusters. We show next that for a well-clusterable dataset $V$, there is a constant $\delta$ that can be computed in terms of the parameters in Definition 1 such that the $\delta$-$k$-means clusters correctly most of the data points.

**Claim A.4.** *Let $V$ be a well-clusterable data matrix. Then, for at least $\lambda N$ data points $v_i$, we have*

$$\min_{j \neq \ell(i)} \left( d^2(v_i, c_j) - d^2(v_i, c_{\ell(i)}) \right) \geq \xi^2 - 2\sqrt{\eta}\beta$$

*which implies that a $\delta$-$k$-means algorithm with any $\delta < \xi^2 - 2\sqrt{\eta}\beta$ will cluster these points correctly.*

*Proof.* By Definition 1, we know that for a well-clusterable dataset $V$, we have that $d(v_i, c_{l(v_i)}) \leq \beta$ for at least $\lambda N$ data points and where $c_{l(v_i)}$ is the centroid closest to $v_i$. Further, the distance between each pair of the $k$ centroids satisfies the bounds $2\sqrt{\eta} \geq d(c_i, c_j) \geq \xi$. By the triangle inequality, we have $d(v_i, c_j) \geq d(c_j, c_{\ell(i)}) - d(v_i, c_{\ell(i)})$. Squaring both sides of the inequality and rearranging,

$$d^2(v_i, c_j) - d^2(v_i, c_{\ell(i)}) \geq d^2(c_j, c_{\ell(i)}) - 2d(c_j, c_{\ell(i)})d(v_i, c_{\ell(i)}))$$

Substituting the bounds on the distances implied by the well-clusterability assumption, we obtain $d^2(v_i, c_j) - d^2(v_i, c_{\ell(i)}) \geq \xi^2 - 2\sqrt{\eta}\beta$. This implies that as long as we pick $\delta < \xi^2 - 2\sqrt{\eta}\beta$, these points are assigned to the correct cluster, since all other centroids are more than $\delta$ further away than the correct centroid.

$\square$

## A.3 Quantum Subroutines

We will assume at some steps that these matrices and $V$ and $C^t$ are stored in suitable QRAM data structures which are described in [23]. To prove our results, we are going to use the following tools:

**Theorem A.5** (Amplitude estimation [7]). *Given a quantum algorithm*

$$A : |0\rangle \rightarrow \sqrt{p} |y, 1\rangle + \sqrt{1-p} |G, 0\rangle$$

*where $|G\rangle$ is some garbage state, then for any positive integer $P$, the amplitude estimation algorithm outputs $\tilde{p}$, with $(0 \leq \tilde{p} \leq 1)$, such that*

$$|\tilde{p} - p| \leq 2\pi \frac{\sqrt{p(1-p)}}{P} + \left(\frac{\pi}{P}\right)^2,$$

*with probability at least $8/\pi^2$. It uses exactly $P$ iterations of the algorithm $A$. If $p = 0$ then $\tilde{p} = 0$ with certainty, and if $p = 1$ and $P$ is even, then $\tilde{p} = 1$ with certainty.*

In addition to amplitude estimation, we will make use of a tool developed in [36] to boost the probability of getting a good estimate for the distances required for the $q$-means algorithm. In high level, we take multiple copies of the estimator from the amplitude estimation procedure, compute the median, and reverse the circuit to get rid of the garbage. Here we provide a theorem with respect to time and not query complexity.

**Theorem A.6** (Median Evaluation [36]). *Let $\mathcal{U}$ be a unitary operation that maps*

$$\mathcal{U} : |0^{\otimes n}\rangle \mapsto \sqrt{a}\,|x, 1\rangle + \sqrt{1-a}\,|G, 0\rangle$$

*for some $1/2 < a \leq 1$ in time $T$. Then there exists a quantum algorithm that, for any $\Delta > 0$ and for any $1/2 < a_0 \leq a$, produces a state $|\Psi\rangle$ such that $\|\,|\Psi\rangle - |0^{\otimes nL}\rangle\,|x\rangle\,\| \leq \sqrt{2\Delta}$ for some integer $L$, in time*

$$2T \left\lceil \frac{\ln(1/\Delta)}{2\left(|a_0| - \frac{1}{2}\right)^2} \right\rceil.$$

We also need some state preparation procedures. These subroutines are needed for encoding vectors in $v_i \in \mathbb{R}^d$ into quantum states $|v_i\rangle$. An efficient state preparation procedure is provided by the QRAM data structures.

**Theorem A.7** (QRAM data structure [23]). *Let $V \in \mathbb{R}^{N \times d}$, there is a data structure to store the rows of $V$ such that,*

1. *The time to insert, update or delete a single entry $v_{ij}$ is $O(\log^2(N))$.*

2. *A quantum algorithm with access to the data structure can perform the following unitaries in time $T = O(\log^2 N)$.*

   (a) *$|i\rangle\,|0\rangle \to |i\rangle\,|v_i\rangle$ for $i \in [N]$.*
   (b) *$|0\rangle \to \sum_{i \in [N]} \|v_i\|\,|i\rangle$.*

In our algorithm we will also use subroutines for quantum linear algebra. For a symmetric matrix $M \in \mathbb{R}^{d \times d}$ with spectral norm $\|M\| = 1$ stored in the QRAM, the running time of these algorithms depends linearly on the condition number $\kappa(M)$ of the matrix, that can be replaced by $\kappa_\tau(M)$, a condition threshold where we keep only the singular values bigger than $\tau$, and the parameter $\mu(M)$, a matrix dependent parameter defined as

$$\mu(M) = \min_{p \in \mathcal{P}}(\|M\|_F, \sqrt{s_{2p}(M)s_{(1-2p)}(M^T)}),$$

where $\mathcal{P} \subset [0, 1]$ is such that $|\mathcal{P}| = O(1)$ and $s_p(M) = \max_{i \in [n]} \sum_{j \in [d]} M_{ij}^p$. The different terms in the minimum in the definition of $\mu(M)$ correspond to different choices for the data structure for storing $M$, as detailed in [22]. Note that $\mu(M) \leq \|M\|_F \leq \sqrt{d}$ as we have assumed that $\|M\| = 1$. The running time also depends logarithmically on the relative error $\epsilon$ of the final outcome state. [8, 18].

**Theorem A.8** (Quantum linear algebra [8] ). *Let $M \in \mathbb{R}^{d \times d}$ such that $\|M\|_2 = 1$ and $x \in \mathbb{R}^d$. Let $\epsilon, \delta > 0$. If $M$ is stored in QRAM data structures with parameter $\mu(M)$ and the time to prepare $|x\rangle$ is $T_x$, then there exist quantum algorithms that with probability at least $1 - 1/poly(d)$ return*

1. *A state $|z\rangle$ such that $\|\,|z\rangle - |Mx\rangle\,\| \leq \epsilon$ in time $\widetilde{O}((\kappa(M)\mu(M) + T_x\kappa(M))\log(1/\epsilon))$.*

2. *A state $|z\rangle$ such that $\|\,|z\rangle - |M^{-1}x\rangle\,\| \leq \epsilon$ in time $\widetilde{O}((\kappa(M)\mu(M) + T_x\kappa(M))\log(1/\epsilon))$.*

3. *Norm estimate $z \in (1 \pm \delta)\,\|Mx\|$, with relative error $\delta$, in time $\widetilde{O}(T_x \frac{\kappa(M)\mu(M)}{\delta}\log(1/\epsilon))$.*

The linear algebra procedures above can also be applied to any rectangular matrix $V \in \mathbb{R}^{N \times d}$ by considering instead the symmetric matrix $\overline{V} = \begin{pmatrix} 0 & V \\ V^T & 0 \end{pmatrix}$.

The final component needed for the $q$-means algorithm is a linear time algorithm for vector state tomography that will be used to recover classical information from the quantum states corresponding to the new centroids in each step. Given a unitary $U$ that produces a quantum state $|x\rangle$, by calling $O(d \log d/\epsilon^2)$ times $U$, the tomography algorithm is able to reconstruct a vector $\widetilde{x}$ that approximates $|x\rangle$ such that $\| |\widetilde{x}\rangle - |x\rangle \| \leq \epsilon$.

**Theorem A.9** (Vector state tomography [24]). *Given access to unitary $U$ such that $U|0\rangle = |x\rangle$ and its controlled version in time $T(U)$, there is a tomography algorithm with time complexity $O(T(U)\frac{d \log d}{\epsilon^2})$ that produces unit vector $\widetilde{x} \in \mathbb{R}^d$ such that $\|\widetilde{x} - x\|_2 \leq \epsilon$ with probability at least $(1 - 1/poly(d))$.*

### A.4 Detailed proofs of the quantum procedures

#### A.4.1 Proof of Theorem 2.1

The theorem will follow easily from the following lemma which computes the square distance or inner product of two vectors.

**Lemma A.10** (Distance / Inner Products Estimation). *Assume for a data matrix $V \in \mathbb{R}^{N \times d}$ and a centroid matrix $C \in \mathbb{R}^{k \times d}$ that the following unitaries $|i\rangle|0\rangle \mapsto |i\rangle|v_i\rangle$, and $|j\rangle|0\rangle \mapsto |j\rangle|c_j\rangle$ can be performed in time $T$ and the norms of the vectors are known. For any $\Delta > 0$ and $\epsilon_1 > 0$, there exists a quantum algorithm that computes in time $\widetilde{O}\left( \frac{\|v_i\|\|c_j\|T \log(1/\Delta)}{\epsilon_1} \right)$,*

$$|i\rangle|j\rangle|0\rangle \mapsto |i\rangle|j\rangle|\overline{d^2(v_i,c_j)}\rangle,$$

*where $|\overline{d^2(v_i,c_j)} - d^2(v_i,c_j)| \leqslant \epsilon_1$ with probability at least $1 - 2\Delta$, or*

$$|i\rangle|j\rangle|0\rangle \mapsto |i\rangle|j\rangle|\overline{(v_i,c_j)}\rangle$$

*where $|\overline{(v_i,c_j)} - (v_i,c_j)| \leqslant \epsilon_1$ with probability at least $1 - 2\Delta$.*

*Proof.* Let us start by describing a procedure to estimate the square $\ell_2$ distance between the normalised vectors $|v_i\rangle$ and $|c_j\rangle$. We start with the initial state

$$|\phi_{ij}\rangle := |i\rangle|j\rangle \frac{1}{\sqrt{2}}(|0\rangle + |1\rangle)|0\rangle$$

Then, we query the state preparation oracle controlled on the third register to perform the mappings $|i\rangle|j\rangle|0\rangle|0\rangle \mapsto |i\rangle|j\rangle|0\rangle|v_i\rangle$ and $|i\rangle|j\rangle|1\rangle|0\rangle \mapsto |i\rangle|j\rangle|1\rangle|c_j\rangle$. The state after this is given by,

$$\frac{1}{\sqrt{2}}\left(|i\rangle|j\rangle|0\rangle|v_i\rangle + |i\rangle|j\rangle|1\rangle|c_j\rangle\right)$$

Finally, we apply an Hadamard gate on the the third register to obtain,

$$|i\rangle|j\rangle\left(\frac{1}{2}|0\rangle(|v_i\rangle + |c_j\rangle) + \frac{1}{2}|1\rangle(|v_i\rangle - |c_j\rangle)\right)$$

The probability of obtaining $|1\rangle$ when the third register is measured is,

$$p_{ij} = \frac{1}{4}(2 - 2\langle v_i|c_j\rangle) = \frac{1}{4}d^2(|v_i\rangle, |c_j\rangle) = \frac{1 - \langle v_i|c_j\rangle}{2}$$

which is proportional to the square distance between the two normalised vectors.

We can rewrite $|1\rangle(|v_i\rangle - |c_j\rangle)$ as $|y_{ij}, 1\rangle$ (by swapping the registers), and hence we have the final mapping

$$A : |i\rangle|j\rangle|0\rangle \mapsto |i\rangle|j\rangle\left(\sqrt{p_{ij}}|y_{ij}, 1\rangle + \sqrt{1 - p_{ij}}|G_{ij}, 0\rangle\right) \qquad (10)$$

where the probability $p_{ij}$ is proportional to the square distance between the normalised vectors and $G_{ij}$ is a garbage state. Note that the running time of $A$ is $T_A = \tilde{O}(T)$.

Now that we know how to apply the transformation described in Equation 9, we can use known techniques to perform the centroid distance estimation as defined in Theorem 2.1 within additive error $\epsilon$ with high probability. The method uses two tools, amplitude estimation, and the median evaluation A.6 from [36].

First, using amplitude estimation (Theorem A.5) with the unitary $A$ defined in Equation 9, we can create a unitary operation that maps

$$\mathcal{U} : |i\rangle \, |j\rangle \, |0\rangle \mapsto |i\rangle \, |j\rangle \left( \sqrt{\alpha} \, |\overline{p_{ij}}, G, 1\rangle + \sqrt{(1-\alpha)} \, |G', 0\rangle \right)$$

where $G, G'$ are garbage registers, $|\overline{p_{ij}} - p_{ij}| \leq \epsilon$ and $\alpha \geq 8/\pi^2$. The unitary $\mathcal{U}$ requires $P$ iterations of $A$ with $P = O(1/\epsilon)$. Amplitude estimation thus takes time $T_{\mathcal{U}} = \widetilde{O}(T/\epsilon)$. We can now apply Theorem A.6 for the unitary $\mathcal{U}$ to obtain a quantum state $|\Psi_{ij}\rangle$ such that,

$$\| \, |\Psi_{ij}\rangle - |0\rangle^{\otimes L} \, |\overline{p_{ij}}, G\rangle \, \|_2 \leq \sqrt{2\Delta}$$

The running time of the procedure is $O(T_{\mathcal{U}} \ln(1/\Delta)) = \widetilde{O}(\frac{T}{\epsilon} \log(1/\Delta))$.

Note that we can easily multiply the value $\overline{p_{ij}}$ by 4 in order to have the estimator of the square distance of the normalised vectors or compute $1 - 2\overline{p_{ij}}$ for the normalized inner product. Last, the garbage state does not cause any problem in calculating the minimum in the next step, after which this step is uncomputed.

The running time of the procedure is thus $O(T_{\mathcal{U}} \ln(1/\Delta)) = O(\frac{T}{\epsilon} \log(1/\Delta))$.

The last step is to show how to estimate the square distance or the inner product of the unnormalised vectors. Since we know the norms of the vectors, we can simply multiply the estimator of the normalised inner product with the product of the two norms to get an estimate for the inner product of the unnormalised vectors and a similar calculation works for the distance. Note that the absolute error $\epsilon$ now becomes $\epsilon \|v_i\| \|c_j\|$ and hence if we want to have in the end an absolute error $\epsilon$ this will incur a factor of $\|v_i\| \|c_j\|$ in the running time. This concludes the proof of the lemma. $\square$

The proof of the theorem follows rather straightforwardly from this lemma. In fact one just needs to apply the above distance estimation procedure from Lemma A.10 $k$ times. Note also that the norms of the centroids are always smaller than the maximum norm of a data point which gives us the factor $\eta$.

### A.4.2 Proofs of Steps 2 and 3

We provide the proof of Claim 2.3:

*Proof.* The $k$-means update rule for the centroids is given by $c_j^{t+1} = \frac{1}{|C_j^t|} \sum_{i \in C_j} v_i$. As the columns of $V^T$ are the vectors $v_i$, this can be rewritten as $c_j^{t+1} = V^T \chi_j^t$. $\square$

Now we provide the proof of the Claim 2.4

*Proof.* We can rewrite $\|c_j - \overline{c_j}\|$ as $\left\| \|c_j\| \, |c_j\rangle - \overline{\|c_j\|} \, |\overline{c_j}\rangle \right\|$. It follows from triangle inequality that the above is inferior to:

$$\left\| \overline{\|c_j\|} \, |\overline{c_j}\rangle - \|c_j\| \, |\overline{c_j}\rangle \right\| + \left\| \|c_j\| \, |\overline{c_j}\rangle - \|c_j\| \, |c_j\rangle \right\|$$

We have the upper bound $\|c_j\| \leq \sqrt{\eta}$. Using the bounds for the error we have from tomography and norm estimation, we can upper bound the first term by $\sqrt{\eta}\epsilon_3$ and the second term by $\sqrt{\eta}\epsilon_4$. The claim follows. $\square$

With the resulting state of after matrix multiplication, by measuring the last register, we can sample from the states $|\chi_j^t\rangle$ for $j \in [k]$, with probability proportional to the size of the cluster. We assume here that all $k$ clusters are non-vanishing, in other words they have size $\Omega(N/k)$. Given the ability to

create the states $|\chi_j^t\rangle$ and given that the matrix $V$ is stored in QRAM, we can now perform quantum matrix multiplication by $V^T$ to recover an approximation of the state $|V^T\chi_j\rangle = |c_j^{t+1}\rangle$ with error $\epsilon_2$, as stated in Theorem A.8. Note that the error $\epsilon_2$ only appears inside a logarithm. The same Theorem allows us to get an estimate of the norm $\left\|V^T\chi_j^t\right\| = \left\|c_j^{t+1}\right\|$ with relative error $\epsilon_3$. For this, we also need an estimate of the size of each cluster, namely the norms $\|\chi_j\|$. We already have this, since the measurements of the last register give us this estimate, and since the number of measurements made is large compared to $k$ (they depend on $d$), the error from this source is negligible compared to other errors.

### A.4.3 Proof of Theorem 3.2

We provide the proof of Theorem 3.2, the main results for $q$-means applied to well-clusterable datasets.

*Proof.* We will use some claims stated and proved in the Supplementary Material, Section A.2. Let $V \in \mathbb{R}^{N \times d}$ be a well-clusterable dataset as in Definition 1. In this case, we know by Claim A.3 that $\kappa(V) = \frac{1}{\sigma_{min}}$ can be replaced by a thresholded condition number $\kappa_\tau(V) = \frac{1}{\tau}$. In practice, this is done by discarding the singular values smaller than a certain threshold during quantum matrix multiplication. By Claim A.2 we know that $\|V\|_F = O(\sqrt{k})$. Therefore we need to pick $\epsilon_\tau$ for a threshold $\tau = \frac{\epsilon_\tau}{\sqrt{k}}\|V\|_F$ such that $\kappa_\tau(V) = O(\frac{1}{\epsilon_\tau})$.

Thresholding the singular values in the matrix multiplication step introduces an additional additive error in $\epsilon_{centroid}$. By Claim A.3 and Claim 2.4 , we have that the error $\epsilon_{centroid}$ in approximating the true centroids becomes $\sqrt{\eta}(\epsilon_3 + \epsilon_4 + \epsilon' + \epsilon_\tau)$ where $\epsilon' = \sqrt{\lambda\beta^2 + (1-\lambda)4\eta}$ is a dataset dependent parameter computed in Claim A.1. We can set $\epsilon_\tau = \epsilon_3 = \epsilon_4 = \epsilon'/3$ to obtain $\epsilon_{centroid} = 2\sqrt{\eta}\epsilon'$.

The definition of the $\delta$-$k$-means update rule requires that $\epsilon_{centroid} \leq \delta/2$. Further, Claim A.4 shows that if the error $\delta$ in the assignment step satsifies $\delta \leq \xi^2 - 2\sqrt{\eta}\beta$, then the $\delta$-$k$-means algorithm finds the corrects clusters. By Definition 1 of a well-clusterable dataset, we can find a suitable constant $\delta$ satisfying both these constraints, namely satisfying

$$4\sqrt{\eta}\sqrt{\lambda\beta^2 + (1-\lambda)4\eta} < \delta < \xi^2 - 2\sqrt{\eta}\beta.$$

Substituting the values $\mu(V) = O(\sqrt{k})$ from Claim A.2, $\kappa(V) = O(\frac{1}{\epsilon_\tau})$ and $\epsilon_\tau = \epsilon_3 = \epsilon_4 = \epsilon'/3 = O(\sqrt{\eta}/\delta)$ in the running time for the general $q$-means algorithm, we obtain that the running time for the $q$-means algorithm on a well-clusterable dataset is $\widetilde{O}\left(k^2 d \frac{\eta^{2.5}}{\delta^3} + k^{2.5}\frac{\eta^2}{\delta^3}\right)$ per iteration.

$\square$

### A.5 Initialization: $q$-means++

We now show that the quantum analogue of the initialization procedure of $k$-means++ can be implemented efficiently using the square distance subroutine estimation for the $q$-means algorithm given in Theorem 2.1. Starting with a random index $j$ we compute the state $\frac{1}{\sqrt{N}}\sum_{i=0}^{N-1}|i\rangle|j\rangle|d^2(v_i, v_j)\rangle$ in time $\tilde{O}(\frac{\eta}{\epsilon_1})$, where $v_j$ is the initial centroid, using our quantum procedure for distance estimation. By applying some arithmetic preprocessing and a controlled rotation we can transfer the distance information as an amplitude to obtain the following state:

$$\frac{1}{\sqrt{N}}\sum_{i=0}^{N-1}|i\rangle|j\rangle|d^2(v_i, v_j)\rangle\left(\frac{d(v_i, v_j)}{2\sqrt{\eta}}|0\rangle + \beta|1\rangle\right)$$

Each square distance has been normalized by $2\sqrt{\eta} \geq max_{i,j}(d(v_i, v_j))$ to be a valid amplitude. Note that postselecting on $|0\rangle$ and measuring the register $|i\rangle$ samples exactly from the probability distribution in the $k$-means++ algorithm as the probability of measuring $(i, 0)$ on second and fourth registers is $\frac{d^2(v_i, v_j)}{4\eta N}$.

We can perform amplitude amplification to boost the probability of measuring $|0\rangle$. For this we need to repeat $O(1/\sqrt{P(0)})$ times the previous steps, with $P(0)$ being the probability of measuring $|0\rangle$. Since $P(0) = \frac{1}{N}\left(\sum \frac{d(v_i,v_j)}{2\sqrt{\eta}}\right)^2$, it is simple to show that $\frac{1}{\sqrt{P(0)}} \leq \frac{2\sqrt{\eta}}{\sqrt{\mathbb{E}(d^2(v_i,v_j))}}$, where $\mathbb{E}(d^2(v_i,v_j))$ is the mean squared distance. Note that for next steps we can use a tensor product of squared distance from previous centroids to compute the minimum distance among them, using Lemma 2.2. In the end we repeat $k-1$ times this circuit, for a total time of $\tilde{O}(k^2 \frac{2\eta^{1.5}}{\epsilon_1\sqrt{\mathbb{E}(d^2(v_i,v_j))}})$. The running time for the $q$-means++ initialization is smaller than that for the $q$-means algorithm, showing than $q$-means++ initialization doesn't cancel any benefit of the $q$-means algorithm. Thus, we can use the $q$-means++ algorithm to provide a speedup compared to the classical $k$-means++.

## A.6 Robust Amplitude Estimation for multiple tomography

Let us make a remark about the ability to use Theorem A.9 to perform tomography in our case. The updated centroids will be recovered in step 4 using the vector state tomography algorithm in Theorem A.9 on the composition of the unitary that prepares $|\psi^t\rangle$ and the unitary that multiplies the first register of $|\psi^t\rangle$ by the matrix $V^T$. The input of the tomography algorithm requires a unitary $U$ such that $U|0\rangle = |x\rangle$ for a fixed quantum state $|x\rangle$. However, the labels $\ell(v_i)$ are not fixed due to errors in distance estimation, hence the composed unitary $U$ as defined above therefore does not produce a fixed pure state $|x\rangle$.

We therefore need a procedure that finds labels $\ell(v_i)$ that are a fixed function of $v_i$ and the centroids $c_j$ for $j \in [k]$. One solution is to change the update rule of the $\delta$-$k$-means algorithm to the following: Let $\ell(v_i) = j$ if $\overline{d(v_i,c_j)} < \overline{d(v_i,c_{j'})} - 2\delta$ for $j' \neq j$ where we discard the points to which no label can be assigned. This assignment rule ensures that if the second register is measured and found to be in state $|j\rangle$, then the first register contains a uniform superposition of points from cluster $j$ that are $\delta$ far from the cluster boundary (and possibly a few points that are $\delta$ close to the cluster boundary). Note that this simulates exactly the $\delta$-$k$-means update rule while discarding some of the data points close to the cluster boundary. The $k$-means centroids are robust under such perturbations, so we expect this assignment rule to produce good results in practice.

A better solution is to use consistent phase estimation instead of the usual phase estimation for the distance estimation step , which can be found in [33, 4]. The distance estimates are generated by the phase estimation algorithm applied to a certain unitary in the amplitude estimation step. The usual phase estimation algorithm does not produce a fixed answer and instead for each eigenvalue $\lambda$ outputs with high probability one of two possible estimates $\overline{\lambda}$ such that $|\lambda - \overline{\lambda}| \leq \epsilon$. Instead, here as in some other applications we need the consistent phase estimation algorithm that with high probability outputs a fixed estimate (that depends on the internal randomness used by the algorithm) such that $|\lambda - \overline{\lambda}| \leq \epsilon$.

We also describe another simple method of getting such consistent phase estimation, which is to combine phase estimation estimates that are obtained for two different precision values. Let us assume that the eigenvalues for the unitary $U$ are $e^{2\pi i\theta_i}$ for $\theta_i \in [0,1]$. First, we perform phase estimation with precision $\frac{1}{N_1}$ where $N_1 = 2^l$ is a power of 2. We repeat this procedure $O(\log N/\theta^2)$ times and output the median estimate. If the value being estimated is $\frac{\lambda+\alpha}{2^l}$ for $\lambda \in \mathbb{Z}$ and $\alpha \in [0,1]$ and $|\alpha-1/2| \geq \theta'$ for an explicit constant $\theta'$ (depending on $\theta$) then with probability at least $1-1/\text{poly}(N)$ the median estimate will be unique and will equal $1/2^l$ times the closest integer to $(\lambda+\alpha)$. In order to also produce a consistent estimate for the eigenvalues for the cases where the above procedure fails, we perform a second phase estimation with precision $2/3N_1$. We repeat this procedure as above for $O(\log N/\theta^2)$ iterations and taking the median estimate. The second procedure fails to produce a consistent estimate only for eigenvalues $\frac{\lambda+\alpha}{2^l}$ for $\lambda \in \mathbb{Z}$ and $\alpha \in [0,1]$ and $|\alpha - 1/3| \leq \theta'$ or $|\alpha - 2/3| \leq \theta'$ for a suitable constant $\theta'$. Since the cases where the two procedures fail are mutually exclusive, one of them succeeds with probability $1 - 1/\text{poly}(N)$. The estimate produced by the phase estimation procedure is therefore deterministic with very high probability. In order to complete this proof sketch, we would have to give explicit values of the constants $\theta$ and $\theta'$ and the success probability, using the known distribution of outcomes for phase estimation.

For what follows, we assume that indeed the state in Equation 2 is almost a deterministic state, meaning that when we repeat the procedure we get the same state with very high probability.

We set the error on the matrix multiplication to be $\epsilon_2 \ll \frac{\epsilon_4^2}{d\log d}$ as we need to call the unitary that builds $c_j^{t+1}$ for $O(\frac{d\log d}{\epsilon_4^2})$ times. We will see that this does not increase the runtime of the algorithm, as the dependence of the runtime for matrix multiplication is logarithmic in the error.

## A.7 Experiment: MNIST dataset details

Table 1: A sample of results collected from the MNIST experiment with PCA preprocessing, same as in Figure 1. Different metrics are presented for the train set and the test set. ACC: accuracy. HOM: homogeneity. COMP: completeness. V-M: v-measure. AMI: adjusted mutual information. ARI: adjusted rand index. RMSEC: Root Mean Square Error of Centroids.

| ALGORITHM | DATASET | ACC | HOM | COMP | V-M | AMI | ARI | RMSEC |
|---|---|---|---|---|---|---|---|---|
| K-MEANS | TRAIN | 0.582 | 0.488 | 0.523 | 0.505 | 0.389 | 0.488 | 0 |
| | TEST | 0.592 | 0.500 | 0.535 | 0.517 | 0.404 | 0.499 | - |
| $\delta$-$k$-MEANS, $\delta = 0.2$ | TRAIN | 0.580 | 0.488 | 0.523 | 0.505 | 0.387 | 0.488 | 0.009 |
| | TEST | 0.591 | 0.499 | 0.535 | 0.516 | 0.404 | 0.498 | - |
| $\delta$-$k$-MEANS, $\delta = 0.3$ | TRAIN | 0.577 | 0.481 | 0.517 | 0.498 | 0.379 | 0.481 | 0.019 |
| | TEST | 0.589 | 0.494 | 0.530 | 0.511 | 0.396 | 0.493 | - |
| $\delta$-$k$-MEANS, $\delta = 0.4$ | TRAIN | 0.573 | 0.464 | 0.526 | 0.493 | 0.377 | 0.464 | 0.020 |
| | TEST | 0.585 | 0.492 | 0.527 | 0.509 | 0.394 | 0.491 | - |
| $\delta$-$k$-MEANS, $\delta = 0.5$ | TRAIN | 0.573 | 0.459 | 0.522 | 0.488 | 0.371 | 0.459 | 0.034 |
| | TEST | 0.584 | 0.487 | 0.523 | 0.505 | 0.389 | 0.487 | - |

In Section 4, we have presented some result concerning a simulation of the $\delta$-$k$-means on the MNIST datsets. Here we detail the results. On top of the accuracy measure (ACC), we also evaluated the performance of $q$-means against many other metrics, reported in Table 1. More detailed information about these metrics can be found in [26, 15]. We introduce a specific measure of error, the Root Mean Square Error of Centroids (RMSEC), which a direct comparison between the centroids predicted by the k-means algorithm and the ones predicted by the $\delta$-$k$-means. Note that this metric can only be applied to the training set. For all these measures, except RMSEC, a bigger value is better. Our simulations show that $\delta$-$k$-means, and thus the $q$-means, even for values of $\delta$ between $0.2 - 0.5$ achieves similar performance to $k$-means, and in most cases the difference is of small magnitude.

## A.8 Additional Experiment: Gaussian clusters dataset

We describe numerical simulations of the $\delta$-$k$-means algorithm on a synthetic dataset made of several clusters formed by random gaussian distributions. These clusters are naturally well suited for clustering by construction, close to what we defined to be a well-clusterable dataset in Definition 1 of Section 1.4. Doing so, we can start by comparing $k$-means and $\delta$-$k$-means algorithms on high accuracy results, even though this may not be the case on real-world datasets.

Without loss of generality, we preprocessed the data so that the minimum norm in the dataset is 1, in which case $\eta = 4$. This is why we defined $\eta$ as a maximum instead of the ratio of the maximum over the minimum which is really the interesting quantity. Note that the running time basically depends on the ratio $\eta/\delta$. We present a simulation where 20.000 points in a feature space of dimension 10 form 4 Gaussian clusters with standard deviation 2.5, that we can see in Figure 2. The condition number of dataset is calculated to be 5.06. We ran $k$-means and $\delta$-$k$-means for 7 different values of $\delta$ to understand when the $\delta$-$k$-means becomes less accurate.

In Figure 3 we can see that until $\eta/\delta = 3$ (for $\delta = 1.2$), the $\delta$-$k$-means algorithm converges on this dataset. We can now make some remarks about the impact of $\delta$ on the efficiency. It seems natural that for small values of $\delta$ both algorithms are equivalent. For higher values of $\delta$, we observed a late start in the evolution of the accuracy, witnessing random assignments for points on the clusters' boundaries. However, the accuracy still reaches 100% in a few more steps.

Figure 2: Representation of 4 Gaussian clusters of 10 dimensions in a 3D space spanned by the first three PCA dimensions.

Figure 3: Accuracy evolution during $k$-means and $\delta$-$k$-means on well-clusterable Gaussians for 5 values of $\delta$. All versions converged to a 100% accuracy in few steps.