[Reviews · NeurIPS 2019]

Reviewer 1



Response to rebuttal: I think the author responses were done well. I think they have satisfactorily answered the questions that I had raised. The reason I am torn between a strong accept and an accept is: most of the techniques used in this paper have already appeared before in various quantum algorithms and are well-known in the quantum community. Having said that, I think putting together known techniques in a rigorous fashion and also practically implementing their algorithm on a quantum simulator is interesting, especially to a problem which is practically important. I think the final aspect might be of interest to a classical ML community; to see how quantum can provide polynomial speedups to relevant ML problems using a toolbag of interesting techniques. I would strongly recommend this paper if there the other referees also back it and there is room for more talks. ------------------- In this paper, the authors look at the problem of k-means clustering: here one is given N d-dimensional vectors v_1,...,v_N in R^d and the goal is to partition the vectors into k classes so that the vectors within a class are close to one another (where closeness is measured as in Euclidean distance). Let V= [ v_1,.., v_N] in R^{N x d}.The complexity of an algorithm for this problem is phrased in terms of the parameters k, d, N, and parameters of V (such as condition number of V, norm of vectors in V). Classically, as far as I can say, Lloyd's algorithm for clustering runs in time O(N k^2 d) per iteration (I am surprised this paper doesn't make a single mention of the classical complexity of this problem). On a high level, in this paper they produce a quantum algorithm that runs linearly in k,d and polylogarithmic in N. In order to phrase their theorem, they consider the following variant of k-means clustering: The introduce delta-k means clustering, where they artifically introduce noise in the label assignment and centroid assignment step (with the motivation that quantum is noisy, which is vague in my opinion). In the delta-k clustering step, instead of choosing the closest centroid to a given point, the algorithm might choose any one of the clusters which is delta-close to it. Additionally, in calculating the centroid, any one of the centroids delta-close to the calculated "optimal centroid" is assigned in the steps of the algorithm. I find these two noisy steps not-so-necessary. In particular, why couldn't just fix delta to be zero and consider a quantum improvement to the standard k-means clustering problem? Is this noise-introducing step really necessary? Final Evaluation: Pros and Cons of this paper: Pros: I think the problem they study is definitely interesting. k-means clustering is definitely a problem that is in the core of classical ML and it deserves attention in the quantum community. Having said that, this submission lacks in various aspects, which I mention below. Cons: There are a number of points which seem missing in this paper. 1) First and foremost, given they want to claim quantum offers an advantage for k-means clustering, I would *strongly* encourage the authors to mention the classical run time of standard clustering algorithms, as well as in the noisy case. 2) They make the assumption that the dataset is well-structured, in this case, does this easen the classical run-time analysis as well? What is the complexity under these assumptions? Recently in quantum machine learning, papers which haven't proven classical lower bounds, in the case of specialized assumptions under which quantum seems to offer an advantage, have been shown to be over-promised. 3) There is a paper by Wiebe, Svore, Kapoor on supervised clustering. On a quick glance, it seems like the techniques used by Wiebe et al.'s paper and this paper are very similar (especially the use of swap test, minimum finding algorithm, "tomography", distance estimation, etc.) and they are solving similar problems. So, how is the algorithm in this paper better than the one in Wiebe et al.s paper which seems to have appeared over 5 years back? 4) Finally, given their submission is to a classical ML conference, I would have liked to see a bit more motivation as to why quantum is important for this problem? Additionally, they seem to consider variants of k-means such as noisy k-means and so on and state these new problems are quantum-amenable, but again it seems like a roundabout way to propose a quantum speed-up for the natural k-means problem. So why consider these variants? Overall, I believe this problem is interesting and deserves attention, but I find this submission lacking in various aspects and wouldn't push for a strong accept. I would accept this paper if only there is space for more quantum talks. More comments on the paper: 1) In lines 66-70, is this practically motivated? In the sense, when implementing practical algorithms for k-means clustering is the noisy step part of the algorithms? 2) I would like more discussion on the QRAM aspect on line 77, are you assuming the same datastructures in ref[20] in your paper as well? If so, what are they, briefly mention them. 3) The scaling of the algorithms in terms of result 1 and 2 are very hairy. I find no intuition as to how one should view kappa(V) and mu(V). I would like some discussion around the results to at least suggest to a reader, how one should view these parameters in terms of datasets. Additionally, I would *strongly* encourage the authors to mention classical upper/lower bounds for these problems in terms of all the parameters, this at least aids the reader in understanding results 1 and 2. 4) In definition 1, do these assumptions also make classical algorithms efficient? 5) Lines 201-207 are slightly confusing, what do you mean measure? It feels like if you choose to measure for the coupon-collector calculation, don't you lose the state each time and that might give a O(k) overhead.

Reviewer 2



Response to rebuttal: I think the rebuttal is well developed and answers most of my questions. I raise some questions in the review because I am not aware of some constraints within the quantum computing framework. The idea proposed in this paper seems to have broad applications in some other EM algorithms, which have high dependency on the data size at each step. But I still hope the authors can state q-means++ more clearly in the formal version. ================================= There has been a lot of interest recently in quantum computing; however, few has used it to solve clustering problems. This paper investigates the problem of applying quantum computing to solve the classic k-means problem, which is known to be NP-hard in most cases. The authors propose a novel quantum algorithm called q-means, and they provide time complexity of their algorithm at each iteration. Since neither quantum simulators nor quantum computers large enough are currently available to test q-means, they argue that a robust delta-k-means algorithm should have similar performance as their algorithm. Some numerical results are presented to show that delta-k-means can achieve almost the same accuracy as plain vanilla k-means. The main strength of the paper lies in the improvement of time complexity per iteration, reducing the linear dependency on data size to poly-log by using quantum information. This is a novel idea and can be used for other clustering algorithms. This paper is well organized except for section 2. Maybe moving the algorithmic scheme in supplementary to paper can help understanding. Some important notations are also omitted.

Reviewer 3



Response to rebuttal: I read the authors' response and I don't have any further comments. -------------------------- ORIGINALITY =========== The proposed q-means algorithm follows along the lines of the classical k-means algorithm but takes advantage of recent advances in quantum computing (such as vector tomography and quantum linear algebra). Its originality lies in putting all the pieces together and giving a rigorous analysis of the resulting algorithm. QUALITY & CLARITY ================= The paper is of a very high quality, and the authors take great care in giving rigorous analyses of all the steps and of performing experiments in order to validate some of the theoretical claims. The writing quality is uneven. In general, the supplementary material is in a much better shape than the 8-page main submission. For instance, in the main text, there's no comparison at all to previous work, or a section which describes the novelty that allows them to beat previous records. SIGNIFICANCE ============ Given the excitement around quantum algorithms for machine learning, this paper is of high significance and should be of interest to many NeurIPS attendees. Also, the quantum parts of the algorithm are nicely packaged into separate modules, so it gives a very good template of how quantum tools can accelerate classical algorithms (provided access to quantum data of course).

[Author Response · NeurIPS 2019]

We thank the reviewers for their insightful comments, and their remarks concerning typos, notation, references. All these will be corrected in the final paper. We answer some of the specific questions raised in the reviews.

**Referee 1**

*On the quantum advantage.* The classical time for a single iteration of $k$-means is $O(kNd)$, we will add this to the main paper. One of the main advantages of $q$-means is to provably reduce this to $O(poly(k, d, \log N))$. Such provable speedups are quite hard to obtain. Recent breakthroughs in quantum-inspired classical algorithms show similar speedups, however these algorithms are strongly impractical (see [arXiv:1905.10415 (2019)] and our Section 5.2, 433-454). Note also that classically, the noisy $k$-means has asymptotically the same running time as $k$-means, in particular the complexity with respect to the number of points remains $O(N)$ and does not drop to $O(\log N)$.

*The well-clusterability assumption.* Like $k$-means, the $q$-means algorithm and its runtime are compatible with any dataset and we do not need the well-clusterability assumption for our analysis. We only consider this assumption at a second stage in order to provide a second, more interpretable, running time for these datasets in terms of $k, d$ rather than $\kappa$ and $\mu$. We do not know of a specialized classical algorithm for well clustered datasets, but a quantum algorithm would probably still offer an advantage due to its logarithmic dependence on the number of data points.

*Comparison with [WKS14].* The $q$-means algorithm is substantially different from [WKS 14], which works on sparse datasets. The first step of $q$-means indeed assigns points to the closest cluster using methods similar to [WKS14]. However, the centroid update step uses quantum linear algebra and tomography techniques that have been developed more recently. This is also the reason that WKS obtains only a quadratic speedup for $k$-nearest neighbors.

*On $\delta$-$k$-means.* The introduction of $\delta$-$k$-means is necessary to have a rigorous classical analogue of our quantum algorithm. Our quantum algorithm has some inherent approximation error due to quantum subroutines like distance estimation and tomography that output estimates with error $\delta$. Therefore a quantum algorithm using these procedures cannot simulate the exact $k$-means. Despite this, $q$-means is still useful for machine learning purposes. Noise is also practically motivated and appears even in some classical implementations to achieve robust clustering and avoid overfitting. This is also confirmed by our experiments with the MNIST and synthetic dataset.

*More comments.* Indeed, the running time of the quantum algorithm is "hairy" but we find it important to include all parameters explicitly, including $\kappa(V)$ (condition number) and $\mu(V)$ (Frobenius norm/spectral norm). A discussion bounding these parameters for well clustered datasets is in section 5.3, and for 'low-rank' data one can think of both parameters as $\sqrt{k}$. For lines 201-207, we measure only the second register, thus the state obtained is one of the $k$ cluster centers almost uniformly at random to which the coupon collector argument can be applied.

**Referee 2**

*Improvement: 1.* A gaussian distribution would indeed improve the empirical performances of the $\delta$-$k$-means algorithm. However, the uniform distribution comes as a consequence of the current quantum algorithm, thus one would need a different variant to work with the gaussian distribution.

*2.* The probability of failure does not affect significantly the running time: for the median it appears within a logarithmic factor; the tomography works with probability $1 - 1/poly(d)$ with the stated running time.

*3.* The distance estimation error is $\epsilon_1$, hence the difference between two distances can err by $2\epsilon_1$, needing $\epsilon_1 = \delta/2$.

*4.* Our work also includes the $q$-means++ algorithm for initialization which follows exactly the $k$-means++ algorithm by creating a probability distribution proportional to the squares of the distances. These distances are indeed estimated with a noise addition of order $\epsilon_1$ that can be tuned. This will be stated more clearly in the main paper.

*5.* The classical $k$-means can be sped up to complexity $O(kdN/p)$ when using $p$ parallel processors, thus one would need an infeasibly large number of parallel processors to match the quantum running time. Moreover, the complexity of the quantum algorithm refers to the size of the circuit and we do not assume parallel processors. Note also, that we could apply our quantum algorithm on each one of these processors and reduce the complexity to $O(kd \log(N/p))$. Quantum algorithms can be more powerful than just parallel algorithms, they involve a completely different set of techniques and can offer in certain cases unrivalled speedups.

*Bottleneck.* Indeed, $k$-means/$q$-means can get stuck in local minima, but $k$-means++/$q$-means++ finds a local optimum that is provably within $\log(k)$ of the true optimal value, and it rarely takes many iterations. In most large data sets, the bottleneck is the cost of each iteration, since one needs to go over all points. This is the great advantage of our algorithm. Even with a considerable number of parallel processors, large data sets still cannot be classically clustered efficiently.

**Referee 3.** We will make our best effort to present clearly all core results, previous work, and ideas within the main paper page limits, and make it easier for the classical ML community to understand the potential of quantum computing.

Finally, some compelling advantages of $q$-means: It is directly comparable to classical algorithms as it has the same input and output. The algorithm is simple enough to be implemented on small quantum computers and could provide an experimental test for quantum machine learning. Despite its flaws, $k$-means is fundamental to classical ML and we think $q$-means will also be fundamental for quantum ML as it provides strong evidence for the relevance of quantum computing for clustering. It is also a great introduction for the classical ML community to the power of quantum, in line with NeurIPS. Last, concerning impact, we note that our techniques can be extended to other problems such as Gaussian Mixture Models, Expectation Maximization, spectral clustering, Neural Networks and such work is under way.

[Meta-Review · NeurIPS 2019]

The reviewers are clear that this paper makes important contributions and may help in drawing the attention of the ML community to the advances in quantum computation (both theoretical and through simulations). Even though mst of the quantum computing tools used by the authors are standard in the Quantum literature, putting them together in a rigorous manner for an important ML problem is a valuable contribution. The author should however take the reviewer comments regarding presentation and style very seriously and incorporate them and the explanation in the author feedback in the camera-ready version. Without that there is a chance that the work will be incomprehensible to a significant chunk of the ML audience and the main purpose of submitting such a paper to an ML venue would be defeated.